# Exploring the Role of the Microbiome in Rheumatoid Arthritis—A Critical Review

**DOI:** 10.3390/microorganisms12071387

**Published:** 2024-07-09

**Authors:** Plamena Ermencheva, Georgi Kotov, Russka Shumnalieva, Tsvetelina Velikova, Simeon Monov

**Affiliations:** 1Clinic of Rheumatology, University Hospital ‘St. Ivan Rilski’, 13 Urvich Str., 1612 Sofia, Bulgaria; p.ermencheva@gmail.com (P.E.); gn_kotov@abv.bg (G.K.); rshumnalieva@yahoo.com (R.S.); doctor_monov@yahoo.com (S.M.); 2Department of Rheumatology, Medical University of Sofia, 13 Urvich Str., 1612 Sofia, Bulgaria; 3Medical Faculty, Sofia University St. Kliment Ohridski, Kozyak 1, 1407 Sofia, Bulgaria

**Keywords:** rheumatoid arthritis, etiopathogenesis, microbiome, microbiota, autoimmunity

## Abstract

Rheumatoid arthritis (RA) is a chronic, autoimmune rheumatic disease characterized by synovial joint inflammation with subsequent destruction as well as systemic manifestation, leading to impaired mobility and impaired quality of life. The etiopathogenesis of RA is still unknown, with genetic, epigenetic and environmental factors (incl. tobacco smoking) contributing to disease susceptibility. The link between genetic factors like “shared epitope alleles” and the development of RA is well known. However, why only some carriers have a break in self-tolerance and develop autoimmunity still needs to be clarified. The presence of autoantibodies in patients’ serum months to years prior to the onset of clinical manifestations of RA has moved the focus to possible epigenetic factors, including environmental triggers that could contribute to the initiation and perpetuation of the inflammatory reaction in RA. Over the past several years, the role of microorganisms at mucosal sites (i.e., microbiome) has emerged as an essential mediator of inflammation in RA. An increasing number of studies have revealed the microbial role in the immunopathogenesis of autoimmune rheumatic diseases. Interaction between the host immune system and microbiota initiates loss of immunological tolerance and autoimmunity. The alteration in microbiome composition, the so-called dysbiosis, is associated with an increasing number of diseases. Immune dysfunction caused by dysbiosis triggers and sustains chronic inflammation. This review aims to provide a critical summary of the literature findings related to the hypothesis of a reciprocal relation between the microbiome and the immune system. Available data from studies reveal the pivotal role of the microbiome in RA pathogenesis.

## 1. Introduction

Rheumatoid arthritis (RA) is a chronic, autoimmune rheumatic disease characterized by synovial joint inflammation with subsequent cartilage and bone destruction, as well as systemic manifestations leading to impaired mobility and reduced quality of life [1]. The etiopathogenesis of RA is still unknown, with genetic, epigenetic and environmental factors contributing to disease susceptibility [2,3]. The link between genetic factors such as “shared epitope alleles” and the development of RA is well known. However, it is still unclear why only part of the carriers suffers a break in self-tolerance and develop autoimmunity. Rheumatoid factor (RF) and antibodies against citrullinated protein/peptides (ACPAs) are known risk factors for the development and severity of RA [4].

RFs are autoantibodies directed against the Fc portion of immunoglobulin G (IgG). Although only RF immunoglobulin M (IgM) is included in the diagnostic criteria for RA [5], other RF isotypes could also be found in the serum of RA patients, including IgG and immunoglobulin A (IgA). The production of transient, polyclonal RF with low affinity has been considered part of the normal immune response to pathogens. In contrast, the production of monoclonal RF with affinity maturation has been linked to RA, and specifically to more aggressive disease with joint destruction and extra-articular manifestations [4]. The higher the titer of RF, the greater the likelihood of an individual developing RA. In addition, higher RF levels are associated with a more severe clinical course and a poorer outcome [6].

ACPAs are autoantibodies with different isotypes directed against several epitopes containing the nonessential amino acid citrulline in proteins. Citrulline is formed by the posttranslational modification of arginine (citrullination/deamination) in the presence of intracellular enzymes—peptidyl arginine deiminases (PADs), thus conferring immunogenic properties on self-proteins [7]. While multiple citrullinated proteins have been discovered in the synovial fluid of RA patients, a fairly small number of ACPAs have been identified, such as those targeting α-enolase, vimentin and fibrinogen [8]. Nevertheless, this so-called RA citrullinome contains a multitude of intra- and extracellular citrullinated proteins. Neutrophils, the most abundant cells in RA synovial fluid, are the major source of intracellular citrullination, while recent findings have also implicated synovial fibroblasts and monocytes [9,10]. A specific cell death cascade in neutrophils initiated by pore-forming mechanisms leading to increased influx of calcium ions has recently been indicated as leukotoxic hypercitrullination (LTH) [11]. LTH may be caused both by host immune mechanisms involving proteins such as perforin, as well as bacteria-derived pore-forming toxins.

Interestingly, *A. actinomycetemcomitans*, a pathogenic strain associated with periodontitis (PD) development, may activate LTH in neutrophils via secretion of such pore-forming toxins. Similar findings have been reported about *Streptomyces* spp., *S. aureus* and *S. pyogenes* [12]. Autoantibodies directed against PADs that enhance their catalytic activity and thus upregulate citrullination have been associated with severe erosive arthritis and pulmonary involvement [13,14]. Smoking is a major environmental factor that increases citrullination in the lung, as evidenced by bronchoalveolar lavage (BAL) studies which reported an increase in PADs and a higher degree of citrullination in BAL cells of smokers compared to healthy individuals [15]. While the presentation of altered citrulline-containing epitopes to immune cells is one possible mechanism for the breach in immune tolerance, a recent paper suggested an additional role of citrullination. In their study, Curran et al. propose that citrullination alters the processing and presentation of autoantigens and creates a so-called ‘citrullination-dependent repertoire’, which stimulates T cells from ACPA-positive patients more robustly than controls [16].

ACPAs show 85–99% specificity for RA and could predict the disease onset in patients with undifferentiated arthritis [17]. The presence of both RF and ACPAs in patients’ serum months to years prior to the onset of clinical manifestations of RA has moved the focus to possible epigenetic factors, including environmental triggers that could contribute to the initiation and perpetuation of the inflammatory reaction in RA [4,17]. Autoimmune diseases have a multifactorial pathogenesis. The interaction between genetic and environmental factors triggers the autoimmune processes [18]. Over the past several years, the role of microorganisms at mucosal sites (i.e., microbiome), including oral mucosa, lung mucosa and gut mucosa, has emerged as an essential mediator of inflammation in RA. Alteration in microbiome diversity takes place early in the disease [19]. Dysbiosis prompts the host immune system dysfunction, resulting in host susceptibility to RA [20]. Multiple mechanisms are considered to explain the role of microbiota dysbiosis as a mediator of systemic inflammation in RA, including molecular mimicry, dysfunction of the intestinal barrier, the influence of metabolites, modulation of T-cell population homeostasis and some genetic factors [21,22].

We aim to review studies exploring the role and potential mechanisms of the microbiome in the development and progression of RA. In addition, this review aims to reveal a novel potential therapeutic and preventive approach to RA management.

## 2. The Microbiome and the Autoimmunity

The term microbiome in humans describes the genomic content of the microorganisms (microbiota) living in a symbiotic way at a particular site in the human body [23]. The microbiome is essential for the development, maturation and function of the central nervous system, including behavior and cognition, as well as the proper functioning of the immune system. A better understanding of the microbiome’s physiological role has led researchers to explore its role in the pathogenesis of autoimmune diseases due to its ability to influence the different aspects of immune reactions, including loss of immune tolerance and autoantibody production.

Moreover, an altered microbiome composition has been linked with the transition from healthy mucosal tissue to a state of immune dysfunction in RA [24]. Dysbiosis at the oral, lung and gut mucosa has been linked to triggering the autoimmune reaction in RA, and infection with some microorganisms has been associated with the risk of developing RA. In particular, *Porphyromonas gingivalis* (*P. gingivalis*), *Mycobacterium tuberculosis* (*M. tuberculosis*), *Mycoplasma* spp. and *Proteus mirabilis* (*P. mirabilis*) have been discussed as specific microorganisms carrying a higher risk of developing RA [25]. The suggested mechanisms through which the microorganisms may drive the onset of RA include molecular mimicry and activation of the immune system through the so-called ‘super-antigens’ [26]. Multiple mechanisms are considered to explain the role of gut microbiota dysbiosis as a mediator of systemic inflammation in RA, except for the abovementioned molecular mimicry: dysfunction of the intestinal barrier, the influence of metabolites derived from gut microbiota, autophagy of IECs, modulation of T-cell population homeostasis and some genetic factors [24].

The possible role of the lung, oral and gut microbiome in the induction of RA and various mechanisms responsible for the production of autoantibodies will be discussed in detail below.

## 3. Lung Microbiome in Rheumatoid Arthritis

Lung disease is a crucial extra-articular manifestation of RA and could manifest with involvement of the airways, the parenchyma, the pleura and the pulmonary vasculature [27]. Demoruelle et al. found underlying lung pathology on high-resolution computer tomography (HRCT) prior to RA onset [28]. Risk factors for lung disease in RA include long-standing disease, environmental factors such as tobacco usage, male gender, as well as the use of medication such as methotrexate and variable association with autoantibody positivity [29]. While some found RF positivity associated with parenchymal involvement/interstitial lung disease (ILD) and ACPA positivity, with airway disease, others found an association between ACPAs and parenchymal lung disease in early RA patients [30]. This leads to the hypothesis that different autoantibodies found in RA may develop in various areas of the lung and the lung mucosa, which are the triggers of deregulated immune response and initiating the autoimmune reaction in RA. There is a direct interaction between gene and inhaled environmental factors (including recognition of pathogen-associated molecular patterns (PAMPs) and damage-associated molecular patterns (DAMPs)) in the mucosal surfaces in the airways, which leads to local accumulation and activation of immune cells and localized airway inflammation.

On the other hand, the lung parenchyma could be exposed to retained inhaled factors and systemic circulating factors, leading to lung inflammation with local immune activation and ACPA production [30]. The hypothesis that the lungs could be the primary site of immune deregulation in RA is supported by the fact that there are increased inflammatory markers in the lungs of patients with early RA compared to healthy controls, as well as the increased citrullination of proteins in the lungs of both healthy smokers and early, untreated ACPA-positive RA patients. ACPAs have been found in the sputum of individuals at risk for developing RA, as well as in the sputum of seronegative RA patients and the BAL in early, untreated ACPA-positive RA patients [31]. Shared citrullinated epitopes have been discovered in the lungs and joints in RA patients, and it has been suggested that immune cells primed in the lungs may elicit their effector function in the joints [32].

A few lung microbiome studies have explored the possible role of microbial–immune cell interaction in driving the local immune response in the lung and mediating the pathogenic autoimmune reaction in RA. Scher et al. found decreased α-diversity of the lung microbiome in early RA patients compared to HCs [33]. The authors found that BAL samples from RA subjects had significantly reduced representation of *Actinomycetaceae* and *Spirochaetaceae* families compared to healthy controls. The abundance of the antifungal commensal microorganism *Pseudonocardia* correlated with higher disease activity and the presence of erosive arthritis. A recent study by Lou et al. found significant differences in microbial type and abundance among patients with ILD secondary to RA and dermatomyositis and healthy controls. The *Prevotella* genus is the most common in BAL of RA subjects [34]. Despite these findings, the role of the lung microbiome in RA remains largely unexplored.

## 4. Oral Microbiome in Rheumatoid Arthritis

PD is a chronic inflammation of tooth-supporting connective tissue, as well as gingiva and alveolar bone, which shares common pathogenic aspects to RA, and an association between the two conditions has been established in the literature [35,36]. *Porphyromonas gingivalis*, a bacterium with a prominent role in PD, has been the focus of multiple studies investigating the connection between PD and RA, mainly due to its potential to generate citrullinated proteins and thus trigger the synthesis of ACPAs and subsequent RA-related immune response [37,38]. The oral microbiome can produce an array of small molecules, some of which share structural homology to self-antigens, which may lead to antibody cross-reactivity and aberrant targeting of host proteins (i.e., molecular mimicry). Again, *P. gingivalis* has been implicated here as its α-enolase protein shares 82% homology to human α-enolase and has been shown to cross-react [39]. Alpha-enolase is consistently identified as an autoantigen [40] in RA patients.

Interestingly, higher levels of antibodies against citrullinated α-enolase (anti-CEP-1) were present in individuals without RA suffering from chronic PD compared to non-PD individuals [41]. The study of Kroese et al. found an increased relative abundance of *Prevotella* in the saliva and *Veillonella* in the saliva and tongue coating in individuals with early RA and those with arthralgia and RA-associated autoantibodies compared to healthy controls [36]. Another PD-associated microorganism, *Aggregatibacter actinomycetemcomitans*, has been shown to initiate a dysregulated activation of certain citrullinating enzymes in neutrophils through the impact of its leukotoxin A, thus causing hypercitrullination [12]. One study of 75 female RA patients found a higher relative abundance of *Prevotellaceae* spp. and *Leptotrichiaceae* spp. and a lower content of butyrate and propionate-producing bacteria. The same study distinguished between patients based on disease activity and found higher levels of *Staphylococcus* spp. in subjects with high disease activity, higher levels of *Porphyromonas* spp. in those with low disease activity and a higher abundance of *Treponema* spp. and *Absconditabacteriales* in those in remission [42]. Increasing evidence has suggested that DNA from PD-associated pathogens has been found in the synovial fluid isolates and induces proinflammatory cytokine production [43,44].

## 5. Gut Microbiome in Rheumatoid Arthritis

An increasing number of studies and clinical trials over the past years have explored and revealed the role of the gut microbiome in the pathogenesis of RA. It is now thought that the gut microbiome is one of the environmental factors playing a crucial role in the development and progression of RA [45].

### 5.1. Gastrointestinal Microbiota and the Immune System Interactions

A diverse community of microorganisms, including bacteria, viruses, archaea, fungi, and protozoa, known as the gut microbiome, resides in the human gastrointestinal (GI) tract. All these microorganisms and their genes and genomes form the gut microbiome. The GI tract has the largest mucosal surface, harboring over 1000 bacterial species and 10^14^ bacterial cells [46]. The development of gut microbiota and the maintenance of their homeostasis is affected by a number of factors, including host factors, the delivery pattern (vaginal or C-section), age, diet, and antibiotics usage. Studies have shown that the modulation of the gut microbiota’s composition is possible using probiotics, prebiotics and fecal microbiota transplantation (FMT) [47].

Hosts may cultivate their gut microbiota by specific and nonspecific factors. The intestinal epithelial cells (IECs) and plasma cells produce substances such as mucus, antimicrobial peptides (AMPs) and immunoglobulin A (IgA), which play the role of microbiota colonization controllers. The mucus, with its two layers, has a critical barrier function and is an adhesive surface for gut microbiota. The outer layer is the binding site [48]. It contains soluble mucins and O-glycans utilized by gut microbiota hydrolases and lyases [49]. The host can ‘choose’ the most appropriate gut microorganisms using a self-defensive mechanism. Structural components and metabolites of the gut microbiota induce Paneth cells’ production of AMPs through a mechanism mediated by receptor activation in a system called microbe-associated molecular patterns (MAMPs) [50]. AMPs protect the host from bacteria, viruses, yeast, fungi, and cancer cells. Cullen et al. found that some human gut microbes can resist high levels of host inflammation-associated AMPs by modifying lipopolysaccharides (LPS) as done by *Bacteroides*, the most abundant Gram-negative genus among gut microbiota. That proves a mechanism of host recognition of commensal and pathogenic species [51]. The local defense of the intestinal mucosa is achieved by producing secretory IgA (sIgA) from plasma cells. sIgA-mediated biofilm formation over the mucosa prevents bacterial invasion of the human body in a process called ‘immune exclusion’ [52].

### 5.2. Genetic Factors That Influence the Gut Microbiota

Host genetic factors contribute to the shaping and maintenance of gut microbiota. In their study, Rawls et al. found that gut habitat selects its community structure. They investigated the reciprocal gut microbiota transplantation in germ-free (GF) zebrafish and mice. The gut microbiota of both receivers after transplantation resembled the microbiota of their conventional species. While the genera in the transplanted community resembled those in the community of origin, their relative abundance changed similarly to the normal gut microbial composition of the recipient host [53,54]. Liu et al. investigated fecal microRNAs (miRNAs) in mice and humans to determine their role as a specific host mechanism of modulating the gut microbiota composition. miRNAs are small, noncoding sequences with a length of 17–27 nucleotides that regulate cell processes in ~30% of mammalian genes by imperfectly binding to the 3′ untranslated region/UTR/of target messenger RNAs (mRNAs), causing either transcription repression or mRNA degradation [55]. In their study, Liu et al. recognized that IECs and homeodomain-only protein (HOP) homeobox gene (Hopx)-positive cells are the two primary sources of fecal miRNAs. They studied dextran sulfate sodium (DSS)-induced colitis and found that the destruction of IECs after treatment with DSS leads to a decrease in fecal miRNAs. Comparing the fecal miRNA profile of GF mice with that of specific pathogen-free (SPF) colonized mice, they reported a higher fecal miRNA abundance in GF mice as opposed to SPF-colonized mice. In addition, miRNA profiles differed between the two populations. These results demonstrate that resident gut microbes affect miRNA expression, and gut microbiota may be modulated by fecal miRNA administration [55].

### 5.3. Gut Microbiota Development during Life

It is generally believed that the development of microbiota begins in utero. A number of studies have recognized bacteria and bacterial DNA in meconium, placenta and amniotic fluid [56]. After birth, rapid microbial intestinal colonization starts. As mentioned above, the delivery pattern also influences the microbiota structure. Vaginally delivered infants have primary gut microbiota dominated by *Lactobacillus* and *Prevotella* originating from the mother’s vaginal microbiota. C-section-delivered infants derive their gut microbiota from the mother’s skin, resulting in the dominance of *Streptococcus*, *Corynebacterium* and *Propionibacterium* [57]. At the age of 3, the composition, diversity and functional capabilities of the infant microbiota become similar to adult gut microbiota [58]. Diet is another factor affecting gut microbiota composition.

In breastfed infants, the species that dominate the gut microbiota are *Lactobacillus* and *Bifidobacterium*. In contrast, the dominant species in infants raised on formula milk are *Enterococcus*, *Enterobacteria*, *Bacteroides*, *Clostiridia* and *Streptococcus* [59]. Exploring the abundance of gut microbiota has improved recently due to sequencing methods. Using 16S ribosomal/rRNA/gene sequencing, it was possible to divide species since the gene is common for all bacteria and archaea and contains nine highly variable regions (V1–V9). Techniques based on whole-genome shotgun metagenomics are more sensitive and have higher resolution [60]. There are two extensive projects aimed at gathering the genetic potential of the human microbial community—METAgenomics of the Human Intestinal Tract/MetaHit/and the Human Microbiome Project/HMP/ [61]. In total, 2172 species have been isolated from individuals and classified into 12 different phyla, of which 93.5% belonged to *Proteobacteria*, *Firmicutes*, *Actinobacteria* and *Bacteroidetes*. The composition and structure of intestinal microbiota also depend on physiological features along the gut. In the small intestine, the dominant flora is the *Lactobacillaceae* family, while anaerobic species from *Prevotellaceae*, *Lachnospiraceae* and *Rikenellaceae* families prevail in the colon [62].

Over the course of one’s lifetime, gut microbiota diversity and homeostasis are affected by the host immunity system and many environmental factors, including geographical location, smoking, surgery, antibiotic usage, toxins and alimentary habits. The alteration in gut microbiome composition and/or diversity, the so-called dysbiosis, is associated with an increasing number of diseases. In recent years, increasing data have been collected from studies with animal and human models showing the microbial role in the immunopathogenesis of autoimmune rheumatic diseases. The host immune system–microbiota interaction is thought to initiate loss of immunological tolerance and autoimmunity. This causes immune dysfunction, and dysbiosis triggers and sustains chronic inflammation [63]. The gut microbiome plays a pivotal role as an environmental factor in the pathogenesis and progression of RA. Many studies with both human and mouse models have shown that the intestinal flora of high-risk individuals indicates gut microbiome alteration before the onset of RA. Therefore, exploring gut microbiome dysbiosis in pre-clinical RA may be a promising approach for preventing RA and potential therapeutic strategies [64].

### 5.4. Immunopathogenesis of RA: Molecular Mimicry

The immunopathogenesis of RA is a complex molecular process. Under the influence of environmental factors, loss of immunological tolerance to self-antigens, autoantibodies production and activation of autoreactive T cells are triggered in genetically predisposed, disease-susceptible individuals. ACPAs specific for RA recognize citrullinated epitopes in a large number of autoantigens, including antigens derived from microbial flora. Therefore, dysbiosis can trigger immune intolerance [65]. Intestinal microbiota influences metabolic homeostasis and the host immune system. Commensal microbiota modulates T-cell subpopulation responses against pathogens. A complex of pattern recognition receptors/PRRs/including toll-like receptors/TLRs/and nucleotide-binding oligomerization domain-like receptors/NLRs/recognize PAMPs, allowing the innate immune system to protect the host from infection. Commensal bacteria have different abilities to interact with PRRs, resulting in proinflammatory or anti-inflammatory responses. Gram-positive and Gram-negative bacteria induce different signal pathways in both innate and adaptive immune cascades [65].

Molecular mimicry has been proposed as a potential pathogenetic process in many autoimmune diseases, including RA [66]. By mimicking autoantigens, intestinal microbial proteins may initiate T-cell-mediated autoimmune reaction with circulating cytokines and autoantibodies production. Pienta et al. investigated two peptides representative of common types of gut bacteria in RA patients. N-acetyl-glucosamine-6-sulfatase (GNS) and filamin A (FLNA) were highly expressed in synovial tissue and joint fluid. They were identified as autoantigens that could provoke T-cell autoreactivity in over 50% of RA patients in comparison with healthy controls. GNS protein has sequence similarities with epitopes from proteins of the *Prevotella* sp. and *Parabacteroides* sp., and FLNA has homologous epitopes with proteins of the *Prevotella* sp. and *Butyricimonas* sp., another gut commensal. Thus, molecular mimicry of GNS and FLNA is a possible mechanism for *Prevotella* to induce RA [66]. Research on individuals with pre-clinical RA has shown homologous epitopes in some intestinal bacteria, including *Citrobacter*, *Bacteroides*, *Eggerthella* and *Clostridium*, with collagen XI and human leukocyte antigen (HLA)-DRB1*0401. Collagen XI is a structural component of joint cartilage but is also used for arthritis induction in DBA/1 mice [67]. The HLA-DRB1 shared epitope is considered to be the main factor for genetic predisposition to RA. Therefore, through molecular mimicry between peptides from some intestinal species and arthritogenic antigens, immune intolerance could be caused in susceptible individuals, resulting in early joint destruction [68]. Maeda Y. et al. investigated the gut microbiota of early RA patients and analyzed T-cell response to arthritis-related autoantigen 60S ribosomal protein LA23a/RPL23a/ [69]. Their study used germ-free SGK-mice, genetically predisposed for RA development, inoculated with feces from an RA patient. Results have shown the domination of the *Prevotellacae* family in the fecal microbiota of RA patients, particularly *Prevoltella copri* sp./*P.copri*/, followed by *Prevotella stercorea* sp. Colonization of SGK mice with RA patients’ fecal microbiota leads to increased CD4+ interleukin-17 (IL-17)-producing T cells and enhanced response to arthritis-related autoantigen RPL23a. Therefore, it has been suggested that *P. copri* may have epitopes mimicking the structure of RPL23A [69]. These data underscore the potentially key role of antigen mimicry by pathogenic microflora in breaking the immune self-tolerance—namely, that antibodies directed against bacterial epitopes which share similar sequences with proteins found in the host synovial tissue and fluid may eventually become ‘autoantibodies’ by targeting genetically similar epitopes in the healthy tissue.

An overview of some alterations in the microbiome in RA is presented in Table 1.

### 5.5. Immunopathogenesis of RA: Microbiota Metabolites

Metabolites derived from intestinal flora represent another potential mechanism supposed to link gut microbiota to RA pathogenesis. These are small molecules produced by the bacterial metabolism of dietary ingredients, modification of host-derived metabolites by gut flora or metabolites synthesized directly by gut bacteria. Immunoregulatory roles of short-chain fatty acids/SCFAs/have been under investigation in multiple studies. The anti-inflammatory effect of SCFAs has been proposed since decreased SCFA levels have been detected in both RA patients and animal models [70]. Luu et al. analyzed the potential immunomodulatory role of pentanoate/valeric acid/in their study. Pentanoate has anti-inflammatory effects reflected in the downregulation of IL-17 production and enhancement of the IL-10 expression in mucosal Breg and CD4+ effector T cells. The concentration of valeric acid depends on the fermentation process of some dietary components. It is shown that *Prevotella* generates predominantly acetate with undetectable levels of pentanoate, which correlates with its proposed role in autoimmune inflammatory diseases [71]. Another SCFA, propionic acid, can activate the GPR41 receptor expressed in dendritic cells and thus suppress the Th2 effector function [72]. Propionic acid has immunoregulatory effects manifested in promoting T regulatory cells/Treg/differentiation and increasing IL-10 levels [68]. Treg polarization and downregulation of proinflammatory cytokines production are modulated by butyric acid. Butyrate is supposed to have a protective effect on RA development by inhibiting autoantibody production [73]. Amino acids have been recognized as gut microbiota-derived metabolites in pre-clinical RA. As precursors of SCFA synthesis, branched amino acids could mediate RA regulation. Yu et al. explored the relationship between the gut microbiome, intestinal metabolites and RA. They profiled the feces of 26 RA patients and 26 healthy controls. They found an increased abundance of *Klebsiella*, *Escherichia* and *Flavobacterium* in RA patients, while in healthy controls, *Fusicatenibacter*, *Megamonas* and *Enterococcus* prevailed. The metabolomic analysis demonstrated depletion in fecal metabolites, including traumatic acid, N-alpha-acetyl-l-lysine, kynurenic acid, 5-Hydroxyindole-3-acetic acid, 3-hydroxy-anthranilic acid in RA patients [74]. Bile acid metabolism appears to have an immunoregulatory role on monocytes, macrophages, dendritic cells and natural killer cells (NK cells) [75]. Zhao et al. found that ascorbate degradation positively correlates with serum levels of proinflammatory cytokines—tumor necrosis factor-α (TNFα) and IL-6. *E. coli* and *S. bovis* were highly associated with ascorbate degradation and linked to arthritis progression. *S. bovis* is suggested to be prevalent in the early stage of RA, while *E. coli* might be essential throughout the entire RA progression [76].

### 5.6. Immunopathogenesis of RA: Intestinal Permeability and Cell Signaling

An impairment of the intestinal barrier and increased intestinal permeability is a potential mechanism for systemic immune responses reported in RA patients [77]. *Collinsella aerofaciens* belongs to species that have also been associated with RA development. The potential to reduce the expression of the tight junction proteins *Collinsela* spp. was demonstrated to increase intestinal permeability in RA murine models. The expansion of *Collinsella aerofaciens* was recognized to cause loss of epithelial barrier integrity and subsequent translocation of exterior antigens across the gut barrier into the host tissue and even circulation, which triggers immune responses in joints [78]. In addition, the serum levels of zonulin, a pre-haptoglobin that increases intestinal permeability and causes leaky gut syndrome, were found to be increased in patients with RA and were also reported in mice with collagen-induced arthritis as a model of RA. In this experimental setting, the upregulation of zonulin and the increased intestinal permeability preceded the onset of arthritis and treatment with zonulin antagonist improved disease symptoms, possibly suggesting that leaky gut syndrome may be an initiating event in the RA cascade [79]. The impaired barrier function of the gut mucosa has also been implicated in the pathogenesis of juvenile idiopathic arthritis (JIA) and ankylosing spondylitis (AS) [80]. Based on these findings, it has been hypothesized that increased intestinal permeability by gut microbiota is another potential mechanism for dysbiosis contribution to RA development [46].

The modulatory effect of gut microbiota on immune cells is a hallmark in the pathophysiology of RA. The mitogen-activated protein kinase (MAPK) and nuclear factor kappa B/NF-κB/signaling pathways are vital transcriptional pathways of RA. Both are mediated by TLRs and activate both the innate and adaptive immune system [81]. Wang et al. investigated the interaction between gut microbiota and CD4+ T cell subpopulations, cytokine levels, and disease activity in RA patients compared to healthy controls [82]. *Firmicutes*, *Fusobacteriota* and *Bacteroidota* were detected in decreased abundance, while Actinobacteria and Proteobacteria were detected in increased abundance in RA patients. *Collinsella* ssp. and *Eggerthella* ssp. belong to the *Actinobacteria* phylum. *Collinsella* enhanced RA activity by modulating the epithelial production of IL-17A. A positive correlation was reported between the abundance of *Megamonas*, *Monoglobus* and *Prevotella* and CD4+ T cell count, as well as cytokine levels. *Megamonas* and *Monoglobus* abundance positively correlated with IL-10 level and Th1 and Th2 counts. Meanwhile, the relative abundance of *Prevotella* and *Monoglobus* positively correlated with the absolute number of Th1 and Th2 cells and IL-4, IL-2, IL-10, TNF-α and IFN-γ levels. RA disease activity strongly correlates with the Treg and Th17/Treg ratio quantity. *Prevotella* genus, especially *Prevotella* copri sp., was recognized as the dominant bacteria in early RA patients. Lipopolysaccharides/LPS/in the outer membrane of Gram-negative *P. copri* are recognized by TLR4 and triggered Th17 cells’ cytokine production, including IL-17, TNF-α, IL-22, IL-21 and granulocyte–macrophage colony-stimulating factor (GM-CSF) [83]. In RA patients, disease activity correlated with the population of Th17 cells and levels of Th17-secreted cytokines [82]. As a mediator of the matrix metalloproteinase (MMP) production by synovial fibroblasts, IL-17 is supposed to enable joint tissue destruction in RA [84].

### 5.7. Immunopathogenesis of RA: Autoantibodies, miRNAs and Microbiome

Intestinal phage communities’ composition correlates with ACPA serology. Autophagy regulates gut microbiota structure and intestinal barrier function. Disturbance of autophagy may alter the gut microbiota homeostasis. Mangalea et al. conducted a study to define intestinal phage populations of anti-CCP-positive and -negative individuals in a cohort at risk for RA. They revealed that the phage community depends on RA-susceptibility status and highlights their potential role as biomarkers for RA progression. Therefore, the interaction between intestinal microbiota, their phages and the host immune system should be considered as a possible mechanism explaining the role of gut microbiome in the etiopathogenesis of RA [85].

A number of data in recent years have revealed that miRNAs interact with host gut microbiome through the regulation of gene expression. miRNAs play an essential role in cell differentiation and immune homeostasis. It was reported that dysregulated miRNA expression affects autoimmune host response, enhances proinflammatory signaling pathways and upregulates the overproduction of proinflammatory cytokines in RA [86,87]. Furthermore, miRNAs influence gut microbiome alteration and impairment of intestinal barrier function. An increased abundance of *Bacteroidaceae* and *Helicobacteraceae* and decreased abundance of *Prevotellaceae*, *Porphyromonadaceae*, *Lachnospiraceae* and *Ruminococcaceae* in deficient IEC miRNA (Dicer1 ΔIEC) mice have been reported. Host miRNAs may enter bacteria, regulate bacterial gene transcription and directly affect bacterial growth [65].

### 5.8. Immunopathogenesis of RA: Role of HLA Alleles and Gut Microbiome

HLA-DRB1 allele has been established as one of the major genetic risk factors for RA [88]. HLA genes interact with intestinal microbiota and modulate host susceptibility to RA. HLA-DRB1 allele is found in more than 70% of RA patients. Studies with murine models of arthritis have demonstrated differences in the fecal microbiome of arthritis-susceptible mice and those resistant to induced arthritis. In their study, Gomez et al. described the fecal microbiome of RA-susceptible HLA-DRB1∗0401 transgenic mice and DRB1∗0402 mice resistant to the development of RA. In the first strain, *Clostridium* sp. was the dominant bacteria, while in the second strain, the families *Porphyromonadaceae* and *Bifidobacteriaceae* were found in greater abundance.

Furthermore, HLA-DRB1*0401 arthritis-susceptible mice have shown a significant increase in intestinal permeability and Th17 cytokine transcription compared to mice carrying the RA-resistant DRB1*0402 gene [89]. Asquith et al. explored the influence of RA-associated HLA-DRB1 allele carriage in healthy individuals over gut microbiome. They found that *Prevotella copri* was strongly positively associated with the HLA-DR allele [90].

As a diverse and complex community of microorganisms, gut microbiota plays a noteworthy role in human health. Altered gut microbiome occurs in RA individuals and differs from healthy controls [80]. Different profiles of intestinal flora have been observed during the course of the disease [91]. *Firmicutes* and *Bacteroidetes* are the dominant phyla in the human gut. At the genera level, *Bacteroides* and *Prevotella* are the prevalent bacteria. An increased abundance of *Prevotella* was discovered in pre-clinical RA patients, while the abundance of *Bacteroides* decreased [69,92]. As already outlined, *Prevotella copri* is the most investigated species of *Prevotella*, playing a pivotal role in the pathogenesis of RA. In contrast, another *Prevotella* sp., *Prevotella histicola*, demonstrates an anti-inflammatory potential and ability to attenuate arthritis through regulation of Treg cellular response and increase in IL-10 transcription [93,94]. Several genera in the phylum *Firmicutes*, including *Lactobacillus*, *Faecalibacterium*, *Streptococcus* and *Blautia*, have been proven to affect RA development and progression. Proliferation of the genus *Lactobacillus* was reported in early RA patients and patients with established disease [95]. Many species of *Lactobacillus* can ease arthritis and reduce inflammation, including *L. reuteri*, *L. casei*, *L. rhamnosus*, *L. acidophilus*, *L. brevis*, *L. salivarius* and *L. fermentum* [96]. Their therapeutic potential may involve different pathways. *L. salivarius* and *L. plantarum* downregulate Th17 cells and enhance Treg cells’ activity. *L. reuteri* and *L. casei* inhibit Th1 immune response, while *L. rhamnosus* and *L. fermentum* impair Th17 responses [97]. Several other bacterial genera have also been associated with human RA, including *Collinsella*, *Eggerthella* and *Actinomyces*. [80]. Both *Eggerthella lenta* and *Collinsella aerofaciens* increase gut permeability and thus contribute to altered immune responses in RA [96]. A decreased abundance of some genera, including *Roseburia* and *Faecalibacterium*, has been demonstrated in RA patients. Producing butyric acid, these bacteria could modulate the inflammatory response through Treg polarization, proinflammatory cytokine downregulation and conventional Th cells’ suppression [98].

### 5.9. Gut Microbiome-Based Interventions in RA

Growing evidence has shown that gut microbiota might exert beneficial and pathogenic effects on human health [65]. Gut microbiome-based interventions could be a promising treatment approach in RA management, as well as in RA prevention. Interventions for treatment and/or prevention of RA through modulation of gut microbiome mainly include probiotics, prebiotics, dietary nutrients, fecal microbiota transplantation and disease-modifying anti-rheumatic drugs (DMARDs). Probiotics such as *Lactobacillus* strains, *Bifidobacterium* spp. could reduce RA symptoms by repairing the intestinal barrier, increasing serum IgA and decreasing the abundance of pathogenic species [99]. Some probiotics can produce antimicrobial compounds like *Lactobacillus reuteri*, which releases reuterin that directly kills harmful microbes [100]. Prebiotics usually consist of nondigestible carbohydrates, oligosaccharides or short polysaccharides like inulin. They activate existing gut microbiota species through increased production of short-chain fatty acids and decreased pH [101]. Fasting therapy and a high-fiber diet can help attenuate RA by improving gut microbiota composition and maintaining intact intestinal barrier function [102].

Fecal microbiota transplantation refers to the process of fecal bacteria transplantation from healthy donors to patients with intestinal dysbiosis to restore the community and function of recipient gut microbiota [103]. Several pathways, such as colonoscopy, enema and enteric-coated capsules, have shown efficiency in FMT [104]. In 2021, Zeng et al. published the first report of FMT use for RA treatment. They showed a good therapeutic effect of FMT in a patient with refractory RA, resulting in reduced arthritis score and RF level [105]. However, additional studies are required to determine the benefits and long- and short-term risks of FMT to restore gut microbiota homeostasis in RA patients.

DMARDs may affect gut microbiota structure and function, while intestinal bacteria could determine the bioavailability and subsequent clinical outcome of DMARDs. For example, methotrexate (MTX) treatment partially restored normal gut microbiota composition in RA patients [106]. However, the role of MTX on intestinal bacteria is not fully beneficial. MTX can affect the conserved bacterial pathways to reduce host immune activation but can also alter gut microbiota diversity in RA patients. This can be compensated by using probiotics or propionate as adjuvant therapy [107]. A recent prospective cohort study found that MTX decreased the subgingival microbial diversity in RA patients but did not impact PD [108]. Another DMARD, sulfasalazine (SSZ), has antibacterial properties. SSZ treatment notably altered the fecal microflora of RA patients by reducing total aerobic bacteria, *Bacteroides* and *E. coli*, and increasing the abundance of *Bacillus* [109]. Treatment with hydroxychloroquine (HCQ) was associated with increased diversity of intestinal flora and suggestive potential for restoration of normal microbiota in RA patients [110]. Etanercept, a TNFα inhibitor, was shown to increase the abundance of *Cyanobacteria* and decrease *Clostridiaceae* in subjects with RA. The authors discussed that *Cyanobacteria* may have anti-inflammatory and immunosuppressant activity owing to the production of secondary metabolites [110]. Adalimumab, another TNFα inhibitor, was found to positively impact the gut microbiome in patients with Crohn’s disease, and a similar mechanism has been suggested in RA [111]. These findings, however, have been somewhat challenged by a recent study by Koh et al., who found that MTX, HCQ and leflunomide had no impact on gut microbiome composition, whereas SSZ decreased bacterial richness.

With regard to biologic DMARDs, the authors found that treatment with TNFα inhibitors increased bacterial diversity and altered the composition of the microbiome compared to treatment with tocilizumab and abatacept. However, the differences were not statistically significant [112]. These conflicting reports indicate that further studies are needed to elucidate how RA therapy may influence the microbiome. In conclusion, exploring the role of the microbiome in RA pathogenesis reveals a novel potential therapeutic and/or preventive approach to RA patients.

Figure 1 presents the complex connection between RA pathogenesis and microbiota, including the tole of environmental factors and therapy.

## 6. Conclusions and Perspectives

The present review has a few important strengths. Herein, we have summarized findings in the recent literature to help better understand the role of the microbiome in RA pathogenesis. Our comprehensive analysis underscores the complex relationship between gut microbiota and the immune system, revealing potential microbial targets for therapeutic intervention. Moreover, we confirm the critical need to integrate microbiome research into the broader context of autoimmune disease management. A potential limitation is the paucity of data regarding the role of microbial species in the immunopathogenesis of seronegative RA, which is an area that still merits additional research.

Furthermore, we have not explored the potential use of antimicrobials in treating RA, which is yet another proof of the role of dysbiosis in initiating and perpetuating the cascade of RA. Exploring the role of the microbiome in RA offers promising opportunities for innovative treatment and management strategies. Microbiome-based diagnostics and personalized medicine could be achieved by elucidating how microbiome alterations influence RA pathogenesis. However, further studies are required to determine the precise mechanistic links between dysbiosis and the development of human RA. Future research should focus on microbiome modulation, which holds the potential to revolutionize RA treatment and significantly improve patient outcomes and quality of life. New directions that merit exploration include the therapeutic potential of targeted probiotic treatment aimed at restoring the composition of the microbiome observed in healthy individuals, as well as the prospect of fecal microbiota transplantation’s use as somewhat ‘etiological’ therapy completely resolving mucosa-driven inflammation in the pathogenic cascade of RA. Additional studies on how RA treatment influences the microbiome and, conversely, how the patient’s bacterial flora alters the therapeutic response could explain why some patients fail to respond to standard therapy and possibly pave the way for personalized treatment choice. Finally, further research into the epigenetic factors involved in altering the structure of the microbiome, including the impact of diet, smoking and exercise, would be essential in the context of lifestyle changes as potential adjuvant steps in the overall management of RA.

## Figures and Tables

**Figure 1 microorganisms-12-01387-f001:**
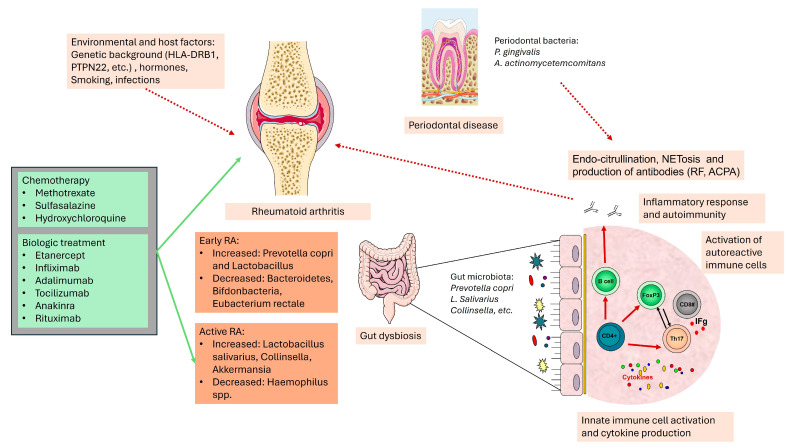
Role of microbiota in rheumatoid arthritis pathogenesis. Invading bacteria can trigger inflammatory (innate immune cell activation, cytokine production, etc.) and autoimmune responses (activation of auto-reactive immune cells, autoantibody production, migration of autoreactive cells to the joints, etc.). Therapy relieves disease symptoms but also improves gut microbiota imbalance. Parts of the figure were drawn by using pictures from Servier Medical Art. Servier Medical Art by Servier is licensed under a Creative Commons BY 4.0 (https://creativecommons.org/licenses/by/4.0/, accessed on 5 July 2024).

**Table 1 microorganisms-12-01387-t001:** Summary of some key alterations in the microbiome associated with RA at different mucosal sites.

Mucosal Site	Microbiome Alterations
Lung mucosa	*Prevotella* spp. are the most common in BAL of RA patients; Presence of *Pseudonocardia* spp. in BAL correlates with higher disease activity and the presence of erosive arthritis; Reduced representation of *Actinomycetaceae* spp. and *Spirochaetaceae* spp. in RA patients vs. healthy controls.
Oral mucosa	*P. gingivalis*, involved in the pathogenesis of PD, has been implicated as a major source of citrullinated peptides (hence, of ACPAs); Possible molecular mimicry by the α-enolase of *P. gingivalis*; Increased *Prevotella* spp. and *Veillonella* spp. in the saliva in individuals with early RA and those with arthralgia and RA-associated autoantibodies compared to healthy controls; Abundance of *Prevotellaceae* spp. and *Leptotrichiaceae* spp. and a lower content of butyrate and propionate-producing bacteria in RA patients; Higher levels of *Staphylococcus* spp. in subjects with high disease activity.
Gut mucosa	*Prevotella* spp. (in particular *P. copri*), *Klebsiella* spp., *Escherichia* spp. and *Flavobacterium* spp. are increased in the feces of RA patients, while *Fusicatenibacter* spp., *Megamonas* spp. and *Enterococcus* spp. predominate in healthy individuals; *Bacteroides* spp. are decreased in the gut microbiome of RA patients; *P. copri* and *S. bovis* are important in early RA, while *E. coli* perpetuates RA progression; *Collinsela* spp. are more abundant in RA patients and are important in increasing the gut mucosal barrier permeability and stimulating RA activity by epithelial production of IL-17A; Proliferation of *Lactobacillus* spp. in patients with early and established RA, likely as a compensatory mechanism due to their ability to suppress inflammation via downregulation of Th17 cells, impairment of the Th1 response and stimulation of Treg cells.

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
