# Peer review of "Exploring the Role of the Microbiome in Rheumatoid Arthritis—A Critical Review"

_microorganisms, 2024, doi:10.3390/microorganisms12071387_

Round 1

Reviewer 1 Report

Comments and Suggestions for Authors

The manuscript by Ermencheva et al. is a review article focusing on the role of the microbiome in rheumatoid arthritis (RA).This review summarizes recent advances in evaluating the potential role of the pulmonary, oral, and intestinal microbiomes in the induction of RA and various mechanisms responsible for the production of autoantibodies.In my opinion, this manuscript is timely, well written, and easy to follow.I have only two minor concerns:

- Why do the authors limit the review to the pulmonary, oral, and intestinal microbiomes? This should be explained.

- In my opinion, there is still potential to improve the presentation by adding more figures/tables summarizing the main ideas of the chapters.

Author Response

Dear Editor,

Thank you for the invitation to review for Microorganisms. I have some comments and suggestions in the above-mentioned manuscript before it can be accepted for publication.

OVERALL COMMENTS

          In this manuscript, the authors found that “Understanding the role of the microbiome in RA offers promising opportunities for innovative treatment and management strategies”. Moreover, they showed the relationship between gut microbiota and the immune system, showing potential microbial targets for therapeutic intervention. Microbiome-based diagnostics and personalized medicine could be achieved by elucidating how microbiome alterations influence RA pathogenesis.

  • RESPONSE: Dear reviewer, thank you very much for your time and efforts to review our paper and provide critical notes for improving our manuscript. We have corrected the issues (visible with track changes).

TITLE

          I suggest including this is a review article.

  • RESPONSE: Thank you for the suggestion. Revised as recommended

ABSTRACT

          Please include the aim of this study in the abstract.

– RESPONSE: Revised as recommended

          I suggest including more information about the findings of this review.

– RESPONSE: Revised as recommended

          In line 17, we can read that “The presence of autoantibodies in patients`sera months to years…” Is this sentence correct?  

- RESPONSE: Patients‘ sera was corrected to Patients‘ serum

KEY-WORDS

 I suggest that the authors reduce the key-words. Live up to six of them.

 – RESPONSE: We reduced the number of keywords as recommended

INTRODUCTION

  1. I miss the aim of the article clearly described.
  2. This section needs some more references. There is a lot of information and only eight references. Moreover, it is necessary to include references published in 2023 and 2024. There are many good articles found in PUBMED.

I wish I could easily find the aim of this study. Please include.

The rest of the text is nice. However, I would ask the authors to include newer references. Please check PUBMED, EMBASE, and Cochrane.

  • RESPONSE: We thank the reviewer for the suggestions. We made the aim clearer and put more references to support our background.

At the end of the manuscript, please include the strengths and limitations of this review.

  • RESPONSE: The strengths and limitations were included as recommended

CONCLUSIONS

          This section is adequate for the findings of the study. I believe that future directions should be expanded.

  • RESPONSE: We are grateful for these points. We have expanded the section covered the future directions in the field.

REFERENCES

          As pointed out above, I suggest including more references published in 2023 and in 2024 in all the sections. 

– RESPONSE: Newer references were included, as referee recommended.  

Reviewer 2 Report

Comments and Suggestions for Authors

Review for the manuscript

Exploring the role of the microbiome in rheumatoid arthritis.

Dear Editor,

Thank you for the invitation to review for Microorganisms. I have some comments and suggestions in the above-mentioned manuscript before it can be accepted for publication.

OVERALL COMMENTS

          In this manuscript, the authors found that “Understanding the role of the microbiome in RA offers promising opportunities for innovative treatment and management strategies”. Moreover, they showed the relationship between gut microbiota and the immune system, showing potential microbial targets for therapeutic intervention. Microbiome-based diagnostics and personalized medicine could be achieved by elucidating how microbiome alterations influence RA pathogenesis.

TITLE

          I suggest including this is a review article.

ABSTRACT

          Please include the aim of this study in the abstract.

          I suggest including more information about the findings of this review.

          In line 17, we can read that “The presence of autoantibodies in patients` sera months to years…” Is this sentence correct?

KEY-WORDS

 I suggest that the authors reduce the key-words. Live up to six of them.

INTRODUCTION

  1. I miss the aim of the article clearly described.
  2. This section needs some more references. There is a lot of information and only eight references. Moreover, it is necessary to include references published in 2023 and 2024. There are many good articles found in PUBMED.

I wish I could easily find the aim of this study. Please include.

The rest of the text is nice. However, I would ask the authors to include newer references. Please check PUBMED, EMBASE, and Cochrane.

At the end of the manuscript, please include the strengths and limitations of this review.

CONCLUSIONS

          This section is adequate for the findings of the study. I believe that future directions should be expanded.

REFERENCES

          As pointed out above, I suggest including more references published in 2023 and in 2024 in all the sections. 

Comments on the Quality of English Language

Minor

Author Response

The comments are addressed in the previous round (Reviewer 1) because this is the same review.

Reviewer 3 Report

Comments and Suggestions for Authors

The manuscript analyzes the link between microbiota, infection, metabolites, and rheumatoid arthritis. Even though the manuscript has a clear rationale, it is not well organized on the basis of the importance of different elements. The role of antigen mimetics should be clarified, and the elements related to citrullination should be better described. The data on oral and gut microbiota should be analyzed along with lung microbiota and skin microbiota. Leaky gut syndrome, especially in children, may lead to juvenile arthritis, for example, doi: 10.3389/fimmu.2021.673708. doi: 10.1093/pcmedi/pbad023

How does arthritis therapy modulate patient's microbiota?

It would be essential to separate in the figure the use of chemotherapy used from biological therapy.

Finally, a perspective on what would be important to analyze in the general population or in patients at high risk of autoimmune disease should be stated.

Comments on the Quality of English Language

Several minor grammatical mistakes were encountered

Author Response

The manuscript analyzes the link between microbiota, infection, metabolites, and rheumatoid arthritis. Even though the manuscript has a clear rationale, it is not well organized on the basis of the importance of different elements. The role of antigen mimetics should be clarified, and the elements related to citrullination should be better described. The data on oral and gut microbiota should be analyzed along with lung microbiota and skin microbiota. Leaky gut syndrome, especially in children, may lead to juvenile arthritis, for example, doi: 10.3389/fimmu.2021.673708. doi: 10.1093/pcmedi/pbad023

– RESPONSE: We thank the referee for their time and efforts to review our paper and provide critical notes for improving our manuscript. We have corrected the issues (visible with track changes) as recommended.

  • The role of antigen or molecular mimicry has been clarified in Section 5.4 and a new paragraph on citrullination has been added in the Introduction. Our thorough search of the pertinent literature revealed that data on skin microbiota are available in the context of psoriatic arthritis but not of rheumatoid arthritis. Leaky gut syndrome has been discussed along with appropriate references.

How does arthritis therapy modulate patient's microbiota?

– RESPONSE: We are grateful for the insightful question. This topic is discussed in the last paragraph before the Conclusion and Perspectives section and new data on biologic DMARD therapy have been added.

It would be essential to separate in the figure the use of chemotherapy used from biological therapy.

  • RESPONSE: We agree with the referee that the two types of therapy are distinct, therefore, we corrected the figure to make this clearer.

Finally, a perspective on what would be important to analyze in the general population or in patients at high risk of autoimmune disease should be stated.

– RESPONSE: Additional perspectives and points of further research have been added at the end of the Conclusion and Perspectives section.

Reviewer 4 Report

Comments and Suggestions for Authors

The authors present a complete review of the state on the art about the role of microbiome in rheumatoid arthritis. The manuscript will be of interest to those working with microbiome and reumathoid arthritis development. 

Just an observation, the manuscript includes a lot of information which provokes in certain degree that becomes difficult to read. In this context, it would be useful to include at least one or two tables with the most important data described in the review.

Finally, a perspectives section would be desirable, something is mentioned at this respect in conclusion section but a deeper description about it would be better.

Author Response

The authors present a complete review of the state on the art about the role of microbiome in rheumatoid arthritis. The manuscript will be of interest to those working with microbiome and reumathoid arthritis development. 

  • RESPONSE: Dear reviewer, thank you very much for your time and efforts to review our paper and provide critical notes for improving our manuscript. We have corrected the issues (visible with track changes).

Just an observation, the manuscript includes a lot of information which provokes in certain degree that becomes difficult to read. In this context, it would be useful to include at least one or two tables with the most important data described in the review.

 – RESPONSE: Thank you for the critical point. We did our best to improve the clarity and readability by separating the information in new passages. A table has been added summarizing the key species in the microbiome of the lung, oral and gut mucosa and their relation to RA

Finally, a perspectives section would be desirable, something is mentioned at this respect in conclusion section but a deeper description about it would be better.

– RESPONSE: Thank you for the valuable insights. Additional perspectives and points of further research have been added at the end of the Conclusion and Perspectives section

Round 2

Reviewer 3 Report

Comments and Suggestions for Authors

The authors have responded most of the queries. There are some points that could have been improved like a better organization of the manuscript 

Comments on the Quality of English Language

Several grammatical mistakes were encountered